# Spectrum of Genetic Variants in the Dystrophin Gene: A Single Centre Retrospective Analysis of 750 Duchenne and Becker Patients from Southern Italy

**DOI:** 10.3390/genes14010214

**Published:** 2023-01-14

**Authors:** Emanuela Viggiano, Esther Picillo, Luigia Passamano, Maria Elena Onore, Giulio Piluso, Marianna Scutifero, Annalaura Torella, Vincenzo Nigro, Luisa Politano

**Affiliations:** 1Department of Prevention, Hygiene and Public Health Service, ASL Roma 2, 00157 Rome, Italy; 2Medical Genetics and Cardiomyology, Department of Precision Medicine, University of Campania “Luigi Vanvitelli”, 80138 Napoli, Italy; 3Telethon Institute of Genetics and Medicine, 80078 Pozzuoli, Italy; 4Cardiomyology and Medical Genetics, Department of Experimental Medicine, University of Campania “Luigi Vanvitelli”, 80138 Napoli, Italy

**Keywords:** *DMD* gene variants, *DMD* gene variants in southern Italy, deletions, duplications, point mutations, gene therapy

## Abstract

Dystrophinopathies are X-linked recessive muscle disorders caused by mutations in the dystrophin (*DMD*) gene that include deletions, duplications, and point mutations. Correct diagnosis is important for providing adequate patient care and family planning, especially at this time when mutation-specific therapies are available. We report a large single-centre study on the spectrum of *DMD* gene variants observed in 750 patients analyzed for suspected Duchenne (DMD) or Becker (BMD) muscular dystrophy, over the past 30 years, at the Cardiomyology and Medical Genetics of the University of Campania. We found 534 (71.21%) large deletions, 73 (9.73%) large duplications, and 112 (14.93%) point mutations, of which 44 (5.9%) were small ins/del causing frame-shifts, 57 (7.6%) nonsense mutations, 8 (1.1%) splice site and 3 (0.4%) intronic mutations, and 31 (4.13%) non mutations. Moreover, we report the prevalence of the different types of mutations in patients with DMD and BMD according to their decade of birth, from 1930 to 2020, and correlate the data to the different techniques used over the years. In the most recent decades, we observed an apparent increase in the prevalence of point mutations, probably due to the use of Next-Generation Sequencing (NGS). In conclusion, in southern Italy, deletions are the most frequent variation observed in DMD and BMD patients followed by point mutations and duplications, as elsewhere in the world. NGS was useful to identify point mutations in cases of strong suspicion of DMD/BMD negative on deletions/duplications analyses. In the era of personalized medicine and availability of new causative therapies, a collective effort is necessary to enable DMD and BMD patients to have timely genetic diagnoses and avoid late implementation of standard of care and late initiation of appropriate treatment.

## 1. Introduction

Duchenne Muscular Dystrophy (DMD) and Becker Muscular Dystrophy (BMD) are progressive degenerative muscle disorders falling in the group of dystrophinopathies, which also includes X-linked Dilated Cardiomyopathy (XL-DCM) and Carrier Dystrophinopathy (CD) [1,2,3]. Dystrophinopathies are caused by mutations in *DMD* gene, which is located on the short arm of the X chromosome at Xp21.1 locus and represents one of the largest genes in humans. The gene encodes a protein called dystrophin, which has a molecular weight of 427 kDa and 3685 amino acids [4,5]. These diseases show heterogeneous clinical phenotypes and genetic characteristics [6,7]. Mutations causing disease include deletions, duplications, and point mutations. The phenotype, BMD or DMD, depends on whether these mutations allow the preservation of the DNA reading frame or not. Usually, out-of-frame mutations result in the total absence of the protein and lead to DMD, the most severe form of dystrophinopathy, while in frame mutations that can determine partial, reduced or localized absence of dystrophin lead to more benign forms [8]. Dystrophin acts as a bridge between sarcolemma and actin cytoskeleton. Though representing only 4% of all muscle proteins, it plays a fundamental role in protecting the membrane from the stress induced by muscle contraction, and in regulating the function of dozens of muscle proteins on the sarcolemma [9]. Dystrophinopathies are inherited as X-linked recessive disorders. In about 2/3 of cases, the transmission occurs through mothers who are carriers of the defective gene, while in the remaining 1/3 of cases the disease arises from *de novo* mutations [10].

### 1.1. Clinical Pictures of Dystrophinopathies

#### 1.1.1. Rapidly Evolving Dystrophinopathy

Duchenne or rather Conte-Duchenne Muscular Dystrophy, from the name of the Neapolitan clinician Gaetano Conte, who in 1836, 32 years before Duchenne [11] described in two brothers what he called “scrofola muscolorum” (muscle degeneration) [12], is the most frequent and more severe muscular dystrophy in children. Its current incidence is estimated at 1:6000 live birth males. The age of onset is between 3 and 5 years, with delay in the motor skills (difficulty in getting up from the ground with “Gowers maneuver” and climbing stairs, frequent falls, waddling gait) [13]. As the disease progresses, muscles of the upper limbs are also compromised. Since adolescence, the heart is also involved in the dystrophic process with evolution towards severe dilated cardiomyopathy [14,15]. The transition to the wheelchair causes the involvement of the respiratory muscles and the progressive decrease in forced vital capacity (FVC) [16]. In the vast majority of cases, the patient’s mental abilities remain unaffected, though the presence of mental retardation or attentional/autism spectrum disorders has been reported in some studies in up to 30% of patients [17]. Creatine kinase (CK) values can increase up to 100 times the maximum normal value [18]. 

#### 1.1.2. Benign Evolving Dystrophinopathy

Becker Muscular Dystrophy (BMD), first described by E. Becker in 1955 [19] shows high phenotypic variability with pictures ranging from asymptomatic hyperCKemia, cramps, and myoglobinuria to mild or moderate muscle involvement characterized by progressive symmetrical muscle weakness and atrophy (proximal greater than distal). Calf hypertrophy is often observed though quadriceps femoral weakness may be the only sign [20,21]. The mean age of onset is approximately 12 years (range 1–70 years). Patients with symptoms and/or signs before age five are indistinguishable from those with DMD. About 50% of patients become symptomatic by the age of 10 and most by the age of 20. However, some patients may not experience muscle symptoms until 50–60 years [22,23]. Respiratory involvement is rare and usually seen in the end stages of the disease. Cardiac involvement is common even before the age of 30 and is always present after this age. In a few patients, heart dilation caused by diffuse fibrosis [15,24], may be the first manifestation of heart involvement and underlying myopathy. Cardiomyopathy is characterized by qualitative defects and/or quantitative abnormalities of dystrophin in the myocardium [24]. Its severity increases with age towards stages of dilated cardiomyopathy and congestive heart failure, passing through a stage of transient compensatory myocardial hypertrophy, sometimes associated with arrhythmias [15,24]. The presence of cardiomyopathy, closely related to the type of dystrophin gene mutation [15,25,26,27,28], strongly influences the life expectancy. Increasingly often, these patients benefit from heart transplantation with excellent long-term results [29,30].

#### 1.1.3. X-Linked Dilated Cardiomyopathy

In XL-DCM [31], dystrophin deficiency is limited to the myocardium. The clinical picture in males between the age of 20 and 50 is characterized by early onset congestive heart failure requiring heart transplantation. However, the disease can similarly affect both boys, with rapid and fatal evolution within 2 years of the onset of symptoms, and older patients. Cardiac involvement, in the form of cardiomegaly or ECG abnormalities is sometimes detected in asymptomatic subjects, during a routine medical examination. Patients may present with signs/symptoms of low cardiac output, or report a history of impaired exercise tolerance and exertional dyspnea in the previous months. The absence of muscle weakness is the rule, but can be associated with increased CK values. Mutations responsible for XL-DCM are located in the 5’ end of the gene and in the spectrin domain [32,33]. Mutations in N-terminal actin binding domain (N-ABD) appear to be associated with a more severe phenotype [32] because the lack of M isoform in the heart is not compensated by alternative mechanisms (upregulation of dystrophin B and P isoforms, exon skipping or alternative splicing), which instead are present in the muscle. Cardiomyopathy in female carriers is slowly progressive.

#### 1.1.4. Carrier’s Dystrophinopathy

Females with dystrophinopathy are usually asymptomatic; however, about 8–10% of them may experience muscle symptoms [34,35,36], while a severe myocardial involvement (16–18% in DMD and 7% in DMB) and in particular dilated cardiomyopathy can occur after the age of 40 [37,38,39].

#### 1.1.5. Treatment of Dystrophinopathies

At present, there are no effective treatments for dystrophinopathies. Glucocorticosteroids (GCS) are regarded as the gold standard of care in DMD. Common practice is treatment with GCS between age 3 and 6 years, until the loss of ambulation, with the aim of delaying cardiac and respiratory involvement and preventing the development of scoliosis. Pharmacological treatment for cardiac manifestations includes the standard treatments of dilated cardiomyopathy and arrhythmias such as the use of angiotensin converting enzyme (ACE) inhibitors, β-blockers and diuretics. Patients with BMD and DMD/BMD carriers may benefit from supportive treatment such as pacemaker (PM) or implantable cardioverter defibrillators (ICD) implant. Heart transplantation is also performed, with excellent long-term results [29,30]. Noninvasive ventilatory support associated with cough assisting techniques, has significantly improved the survival in DMD. In the last years, an increasing number of strategies were studied aimed to ameliorate the DMD phenotype by restoring dystrophin production and preserving muscle mass. Therapeutic approaches that aim to restore partially functioning muscle dystrophinin patients with DMD focus on one of these three approaches: 1. Gene delivery using viral vectors [40]; 2. Stop codon read-through [41]; 3. Converting out-of-frame mutations to in-frame mutations (exon skipping, multiple approaches) [42].

The purpose of this large single-centre retrospective study was to report the spectrum of *DMD* gene variants observed in 750 patients from southern Italy with suspicion of dystrophinopathy, in the last 30 years. As a second step, to report the prevalence of the different types of mutations in patients with DMD and BMD according to their decade of birth, from 1930 to 2020, by correlating the data to the different techniques used over the years.

## 2. Patients and Methods

### 2.1. Patients

A systematic full DMD gene mutation analysis was carried out at the Cardiomyology and Medical Genetics of Naples University (now Luigi Vanvitelli Campania University) Hospital in the period 1991–2021 in patients with the suspicion of muscular dystrophy. They prevalently came from southern Italy (Abruzzo, Apulia, Basilicata, Calabria, Campania and Molise) and Sicily. The informed consent for genetic analysis was obtained from patients or parents/guardians if minors, before blood sampling at the time of the first visit, as per established practice in the University Hospital. Only Duchenne and Becker patients still alive at the time in which genetic analysis was available were included in the study. 

### 2.2. Genomic Analysis

Several different techniques were used to perform the genetic analysis, depending on the period in which they were carried out. In the 1990s, the Southern blotting (SB) technique was the gold standard for detecting deletions, replaced by PCR and quantitative PCR [43,44,45] as they became available. Currently, multiplex ligation-dependent probe amplification (MLPA) analysis is the most reliable test to identify deletions or duplications [46,47,48], also due to its cost- and labor-efficiency. Since 2014, Next-Generation Sequencing (NGS) was introduced as the prominent strategy [49,50,51], to provide a reliable genetic diagnosis in patients who were negative to MLPA analysis. Overall, targeted NGS can bolster a more precise understanding of ambiguous mutations [49,50], and identify pathogenic small mutations also in DMD/BMD patients without a large deletion/duplication, especially in non-coding regions [51,52].

## 3. Results

### 3.1. Distribution of Dystrophin Gene Pathogenic Variants in the Study Population

A pathogenic variant in the dystrophin gene was detected in 719 (95.87%) patients, 467 with the Duchenne phenotype and 252 with the Becker phenotype. Taken together, 534 (71.21%) of them had large deletions, 73 (9.73%) large duplications and 112 (14.93%) point mutations. No mutation was found in 31/750 patients (4.13%) (Figure 1).

Figure 2 shows the distribution of the type of mutations in Duchenne (Figure 2A) and Becker (Figure 2B) patients according to their decade of birth.

The most frequent hot spot of deletions was between E45–E55 (Figure 3A), while the most frequent hot spots of duplications involved exons E2–E23, and less frequently, exons E41–E60 (Figure 3B).

Point mutations were identified in 112 (14.9%) patients, without preferential hotspots (Figure 4). They were distributed as follows: 44 (5.9%) were small ins/del causing a frame-shift, 57 (7.6%) were nonsense mutations, 8 (1.1%) were splice sites, and 3 (0.4%) were intronic mutations (Figure 5).

### 3.2. Dystrophin Gene Variant Analysis in Duchenne Population

Three-hundred-fourteen DMD patients (68.1%) showed large deletions. The groups of exons most frequently deleted were E48–E50 (6.4%), E45–E50 and E46–E47 (5.1%), followed by E46–E48 (4.1%) and E49–E50 (3.8%). The largest deletions involved the entire dystrophin gene in one patient, 35 (E9–E43; E10–E44), 34 (E10–E43), or 30 (E3–E32) exons, respectively, in another four patients. Deletions of only one exon occurred in 80 patients (25.47%), the most frequent of them involved the exons E51 (63.7%), E44 (17.5%) or E45 (13.75%), followed by exons E52 (10.0%) or E50 (10.9%). Exon deletions prevalently occurred in one hot spot, at the 3′ end (exons E44–E55) of the DMD gene (Figure 6).

Fifty-five DMD patients (11.8%) had large duplications, the largest of them involving 37 exons (E33–E60). Eighteen (32.7%) patients presented a single exon duplication, which involved the exons E2, E12, E44, E50, E51, E52, E54, or E56.

The exons most frequently duplicated were E2 and E44 in 27.8% and 22.2% of cases, respectively. Duplications prevalently occurred at the 5′end in the exons E2–E23 and with a minor frequency at the 3′ end, in exons E44–E60 of the *DMD* gene (Figure 7).

Ninety-eight DMD patients (20.98%) showed point mutations, which were randomly distributed along the *DMD* gene, without preferential hotspots.

### 3.3. Dystrophin Gene Variant Analysis in the Becker Population

Two-hundred-twenty (87.3%) BMD patients had large deletions. The groups of exons most frequently deleted were E45–E47 (15.9%), E45–E48 (13.2%) or E45–E49 (10.9%). The deletion of these three groups of exons plus the deletion of exon 48 alone (7.3.%) accounts for about half (47.3%) of all deletions in patients with BMD. The largest deletions involved 43 (E17–E59) or 11 exons (E45–E55). Deletions of a single exon were found in 25 patients (11.4%), and beyond exon E48, they involved exons E13 or E30. Deletions occurred in two hot spots, at the 5′ end (E2–E11) and 3′ end (E45–E55) of the *DMD* gene (Figure 8).

Eighteen BMD patients (6.7%) had large duplications, the largest of which involved 15 exons (E31–E44). Four patients (22.2%) had single exon duplications, which involved exons E2 (two patients) or E44 (two patients). The preferential hot spots of duplications were in exons E13–E20 at the 5′end, and in exons E41–E44 at the 3′end of the gene (Figure 9).

Fourteen patients with BMD (5.2%) had point mutations that were localized in exons E29 (3), E70 (1) or in introns I62 (9) or I75 (1). The pathogenic variant located in intron 62 affected nine members of the same family, in three generations.

## 4. Discussion

Dystrophinopathies are X-linked progressive muscle disorders caused by mutations in the *DMD* gene. The mutations causing the diseases include deletions, duplications, and point mutations.

Having a correct diagnosis of DMD/BMD is important for providing proper care to patients and for family planning, but also for assessing whether patients are eligible for mutation-specific treatments [53].

In this retrospective study, we report the genetic profile of 719 DMD/BMD patients from southern Italy (Abruzzo, Apulia, Basilicata, Campania, Calabria, Molise) and Sicily, detected in the last 30 years in the same laboratory with several techniques.

In our laboratory, the genetic testing for mutations in the dystrophin gene began in the second half of 1991, using the SB analysis and the multiplexes PCR of Beggs [43] and Chamberlain [44], which covered at that time about 98% of all deletions. In the following years, the *DMD* genetic approach in our laboratory evolved according to the progress of techniques, passing from SB and PCR to the MLPA technique [46,47,48] and the more recent Next-Generation Sequencing (NGS) technology [49,50,51,52], obtaining an increase in the detection rate results [54,55,56,57,58,59,60]. This was also possible because the Naples Human Mutation gene Biobank (NHMB)—currently a member of the EuroBioBank (EBB) [61] and the Telethon Network of Genetic Biobanks (TNGB) [62]—were established in our service in 1991, allowing the storage of patients’ blood samples and the re-analysis with NGS of patients’ DNA samples that were negative using previous techniques.

Despite these improvements, a diagnostic delay of up to 4–5 years between the clinical presentation of symptoms and molecular analysis, is still reported in literature. In Italy, the mean age at first medical contact, which raises the suspicion of a DMD, is 31 months, while the mean age at molecular diagnosis is 41 months, 10 months later. In southern Italy, the time for delayed diagnosis is 9 months [63].

We observed over time, in our population, a decrease in the number of patients affected by DMD/BMD, with a parallel reduction in the percentage of point mutations, especially in the last decade. This trend may be the result of genetic counseling and prenatal diagnosis offered to all women at risk of carrying a *DMD* gene defect, or the fact that patients without *DMD* gene deletions/duplications arrive later in tertiary centers and labs, and that patients with BMD are usually diagnosed after age 12, regardless of the type of the molecular variation. However, while genetic counseling and prenatal diagnosis of affected males will certainly contribute to a steady reduction in the number of familial cases in the coming years, no strategy will be able to prevent the occurrence of new cases due to “de novo” mutations, which represent about 1/3 of the total.

Deletions represent the highest percentage of mutations in our patients, followed by point mutations, mainly nonsense variants, in patients with DMD and by duplications in patients with BMD. However, when comparing our percentages with those previously reported by Neri et al. in the Italian population [64], we found that point mutations are overestimated in that study (24.6% versus 14.9% in this study). This difference may lie in several factors, such as the origin of the patients (from all over Italy rather than limited to southern Italy), the number of patients analyzed, or the period of analysis. Since deletions (and duplications) are variants more easily identifiable even with less sophisticated techniques, we believe that the sample reported in Neri’s article was particularly enriched by the enrolment (or referral) of patients for whom no deletions/duplications were detected in the peripheral laboratories. An overestimation of point mutations has also been reported by Flanigan et al. [65], who similarly explain the finding of such a large number of point mutations in their study.

Our data are in agreement with previous studies reporting the frequency of mutations in the *DMD* gene, according to geographical areas in the world (Table 1, and references herein). We considered seven macro-areas—Western and East Europe, Asia, North and South Africa, North and Central-South America—that include 40 Countries. 

The differences in the frequency of the different type of mutations observed between our cohort of patients and other parts of the world, but also among the seven considered macro-areas may depend on several factors, such as: -selection bias, as some studies analyzed subjects from different institutions [79];-inclusion in the study of a high number of patients still undiagnosed and/or negative when using standard techniques [52,68,71,79];-failure to sequence all patients negative for deletions or duplications, due to the cost problem [93];-relatively recent availability of NGS [49,51,52,66];-genetic or environmental predisposition [48,79];-multi-ethnicity of some countries [105,106];-higher rate of inbreeding in some countries, such as the Middle East and Turkey, leading to a higher incidence of autosomal recessive diseases;-limitations of the techniques used, which could explain the low frequency of point mutations in some studies [48,75].

The average results from the seven macro-areas are summarized in Table 2. Overall, a total of 25,705 individuals with the suspicion of dystrophinopathies were analyzed. 

The highest number of DMD/BMD patients were investigated in Asia and Europe, while only 1/4 of the patients analyzed in Europe were analyzed in North America, despite a population 25% higher than Europe. This finding is most probably due to a complex mix of different factors, such as:-the geographic size of North America, which limits the access to diagnostic procedures;-the large difference in urban and rural medical care;-the lack of knowledge in primary care providers about DMD, resulting in less CK and transaminase testing;-insurance coverage issues, especially for genetic testing procedures.

In this study—the largest, by number of patients, performed to our knowledge in a single-center and over such a long period—the detection rate of *DMD* gene variations is 95.8%, the highest among the seven macro-areas. This result was possible because, in our lab, we usually re-analyze all patients with suspected dystrophinopathy, who were negative to previous techniques, using new generation techniques.

It is intriguing to note that dystrophinopathies, with their higher frequency of deletions, differ from other genetic diseases, in which point mutations are the most frequent pathogenic variants (http://www.hgmd.org (accessed on 6 November 2022)), and among them missense variants usually prevail [113]. A possible explanation for the high percentage of deletions could rely on the giant size of the DMD gene and/or on the presence of numerous repeat sequences, that can favor deleterious crossover events during meiosis. The low percentage of point mutations, on the other hand, may be related to the difficulty of their detection using standard techniques, or to the fact that they may cause non-severe (and therefore undiagnosed) phenotypes, leading, consequently, to their underestimation. 

## 5. Conclusions

The life expectancy of patients with DMD has doubled, and more and more patients reach and exceed the fourth decade of life [114] thanks to a multifaceted approach that makes use of pharmacological treatment, cardiac and respiratory support, and rehabilitation.

Therapeutic approaches that aim to restore partially functional muscle dystrophin in patients with DMD are now available for patients with deletions of one or more exons [41], or with stop-codon (nonsense) point mutations [42]. However, the possibility to have access to genetic testing for DMD is not homogeneous in the different countries of the world, causing not only a delay in diagnosis and timely treatment, but also a delay in genetic counseling, implementation of standard care, and participation in clinical trials. 

In this study, we found that the major hot spot for deletions is in exons E45–E55, which are currently the most suitable for gene therapy via exon skipping [40,41].

The data reported here also show that, in industrialized countries, the evolution of molecular strategies was able to improve the detection rate of *DMD* gene variants and the achievement of an early and precise genetic diagnosis in patients with BMD/DMD. 

In the era of personalized medicine and the recent availability of causative therapies, a collective effort is necessary to enable patients with DMD or BMD to have timely genetic diagnoses, even in economically disadvantaged countries, to avoid delayed genetic counseling and problems in family planning, late implementation of the standard of care, and late initiation of treatment.

## Figures and Tables

**Figure 1 genes-14-00214-f001:**
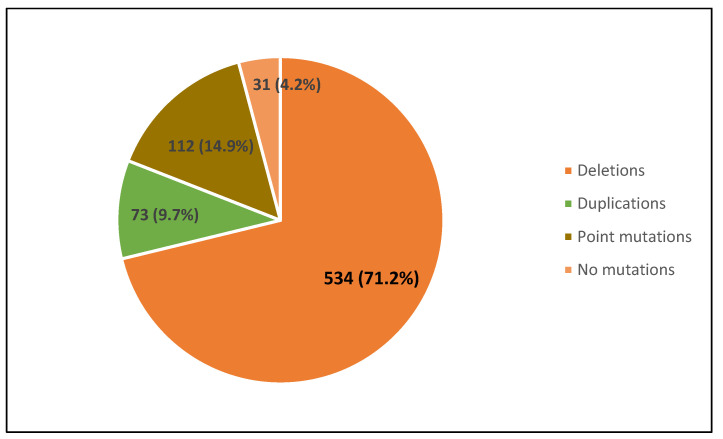
Distribution of the pathogenic *DMD* variants identified in the study population.

**Figure 2 genes-14-00214-f002:**
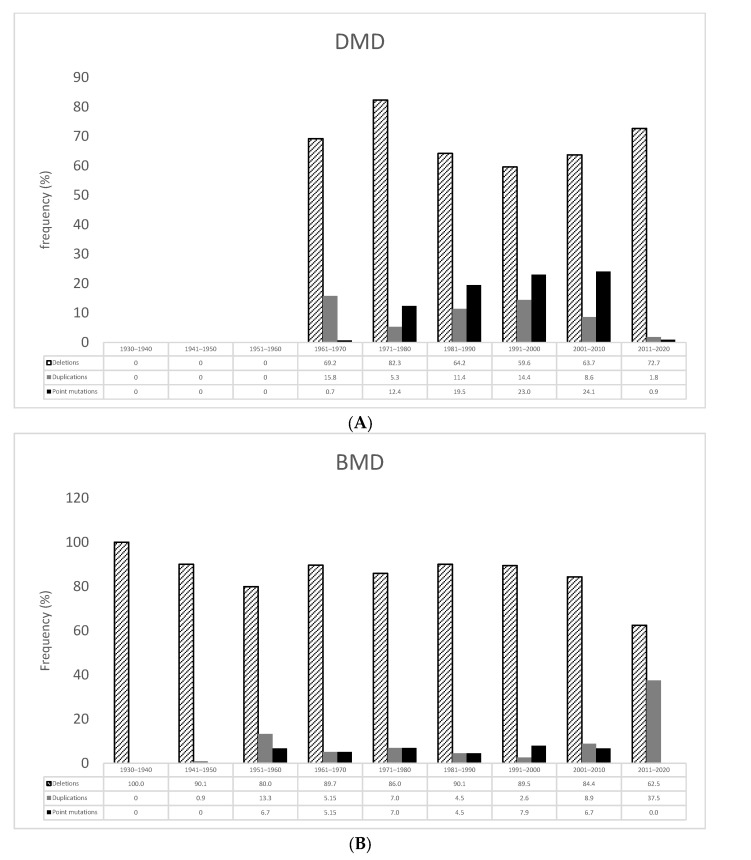
Frequency of mutations in DMD (**A**) or BMD (**B**) analyzed patients according to their decade of birth.

**Figure 3 genes-14-00214-f003:**
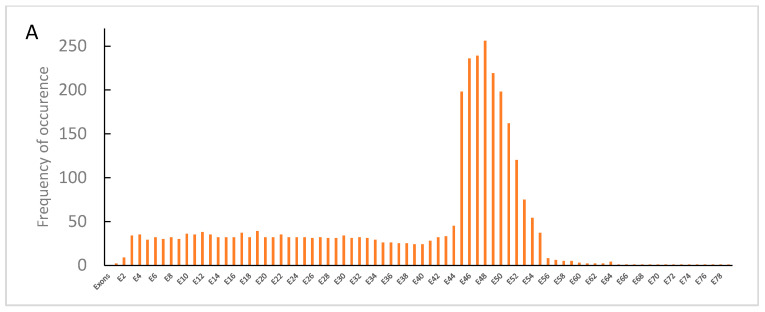
Number of times each individual exon of the *DMD* gene was involved in a deletion (**A**) or duplications (**B**) in the 719 patients with DMD or BMD.

**Figure 4 genes-14-00214-f004:**
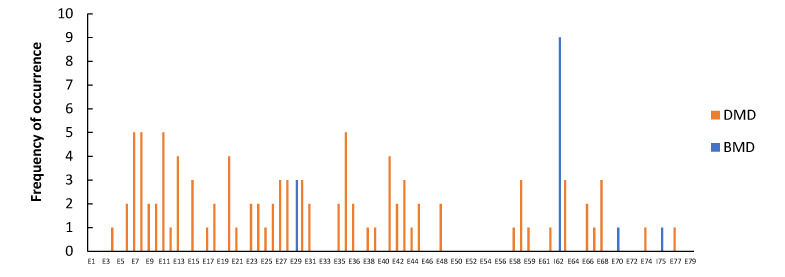
Distribution of point mutations in DMD (red) or BMD (blue) analyzed patients.

**Figure 5 genes-14-00214-f005:**
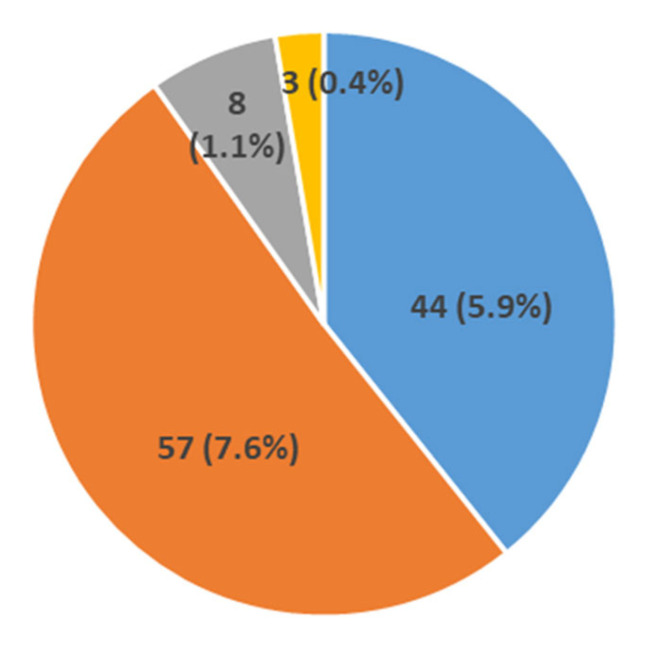
Distribution of the *DMD* gene sub-type point mutations in the analyzed patients with DMD or BMD.

**Figure 6 genes-14-00214-f006:**
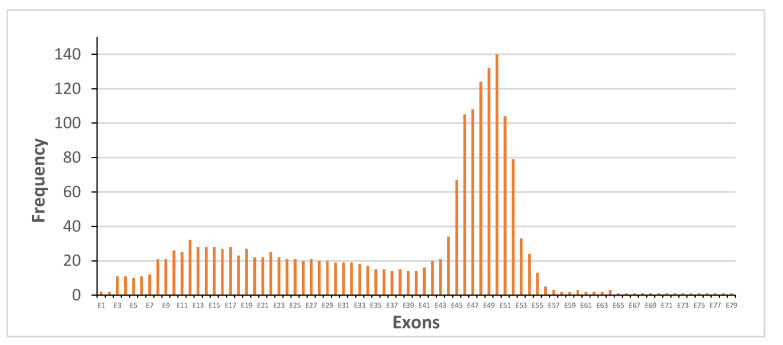
Frequency of single exon deletions in the dystrophin gene, observed in patients with DMD.

**Figure 7 genes-14-00214-f007:**
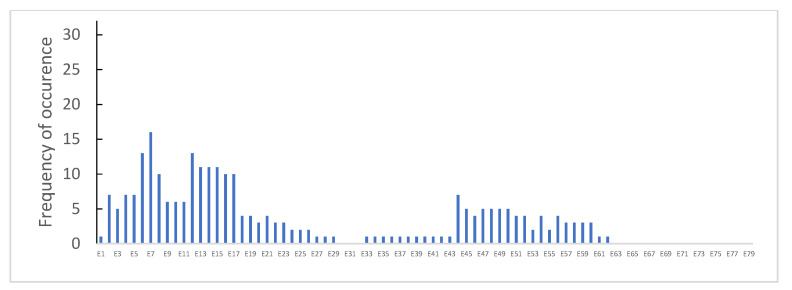
Frequency of each single exon duplication in the dystrophin gene observed in patients with DMD.

**Figure 8 genes-14-00214-f008:**
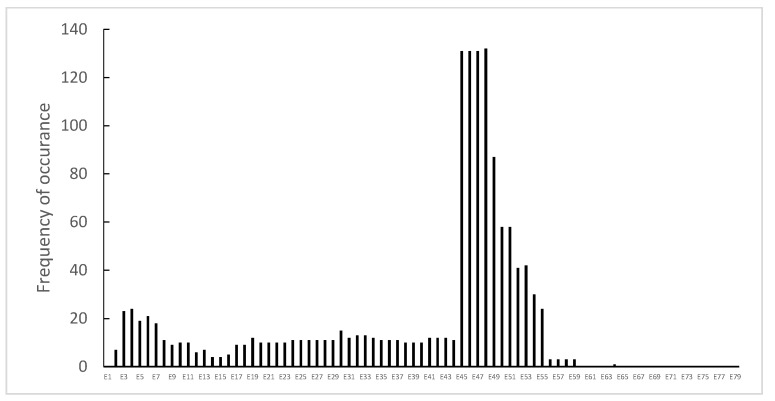
Frequency of single exon deletions in the dystrophin gene observed in patients with BMD.

**Figure 9 genes-14-00214-f009:**
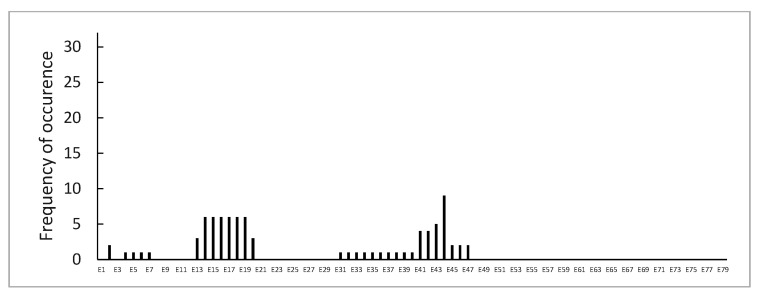
Frequency of single exon duplications in the dystrophin gene observed in BMD patients.

**Table 1 genes-14-00214-t001:** Distribution of dystrophin gene pathogenic variants in the different countries of the world, based on their geographical area. The countries were subdivided into seven macro-areas: Western Europe, East Europe, Asia, North Africa, South Africa, North America, and Central-South America.

Countries	Suspected Dystrophinopathies	Large Deletions		Large Duplications		Point Mutations/Small Rearrangments		Small Ins/Del		Nonsense		Frame-Shift		Splice Site		Missense		Reference
	n.	n.	%	n.	%	n.	%	n.	%	n.	%	n.	%	n.	%	n.	%	
EUROPE																		
Denmark	182	87	47.8	14	7.7	4	2.2	3	1.6	1	n.r.	n.r.	n.r.	n.r.	n.r.	n.r.	n.r.	[48]
France	2411	1404	58.2	362	25.6	465	19.2	n.r.	n.r.	187	7.6	148	6.1	125	8.8	7	0.4	[66]
France	2898	1901	65.5	323	11.1	633	21.8	201	6.9	254	8.7	n.r.	n.r.	158	5.4	20	0.7	[67]
Spain	284	131	46.1	56	19.7	97	34.1	34	12.0	49	17.2	n.r.	n.r.	10	3.5	2	0.7	[68]
Spain	53	34	64.2	4	7.5	n.r.	n.r.	n.r.	n.r.	n.r.	n.r.	n.r.	n.r.	n.r.	n.r.	n.r.	n.r.	[69]
Netherlands	462	212	45.8	42	9.0	n.r.	n.r.	n.r.	n.r.	49	10.6	n.r.	n.r.	18	3.8	n.r.	n.r.	[70]
Italy	1902	1242	65.2	190	9.9	469	24.6	n.r.	n.r.	200	10.5	139	11.1	67	5.4	38	1.9	[64]
**South Italy** **and Sicily**	**750**	**534**	**71.2**	**73**	**9.7**	**112**	**14.9**	**44 ***	**5.9**	**57**	**7.6**	**44 ***	**5.9**	**8**	**1.1**	**0**	**0**	**This study**
EAST EUROPE																		
Bosnia, Bulgaria, Cyprus, Croatia, Hungary, Lithuania, Poland, Romania, Russia, Serbia, and Ukraine	260	52	20	17	6.5	n.r.	n.r.	n.r.	n.r.	49	18.8	29	11.1	16	9	4	1.5	[71]
Poland	180	110	61.1	21	4.7	2	1.1	n.r.	n.r.	1	0.5	1	0.5	n.r.	n.r.	n.r.	n.r.	[72]
Turkey	260	128	49.2	25	9.6	60	23	n.r.	n.r.	32	12.3	17	6.5	1	0.3	4	1.5	[73]
Czech Republic and Slovakia	126	77	61.1	12	9.5	18	14.3	n.r.	n.r.	n.r.	n.r.	n.r.	n.r.	n.r.	n.r.	n.r.	n.r.	[74]
ASIA																		
Saudi Arabia	45	21	46.7	8	17.8	n.r.	n.r.	n.r.	n.r.	n.r.	n.r.	n.r.	n.r.	n.r.	n.r.	n.r.	n.r.	[75]
Saudi Arabia	15	6	40	n.r.	n.r.	n.r.	n.r.	n.r.	n.r.	n.r.	n.r.	n.r.	n.r.	n.r.	n.r.	n.r.	n.r.	[76]
Kuwait	68	45	66.1	3	4.4	6	8.8	n.r.	n.r.	4	5.8	n.r.	n.r.	n.r.	n.r.	2	2.9	[77]
China	1051	740	70.4	87	8.2	201	19.1	58	5.5	102	9.7	n.r.	n.r.	30	2.8	15	1.4	[78]
China	67	23	41.8	5	7.5	23	34.3	6	8.9	13	19.4	n.r.	n.r.	4	5.9	n.r.	n.r.	[79]
China	613	360	60.2	59	9.6	143	23.3	52	8.4	70	11.4	n.r.	n.r.	21	3.4	n.r.	n.r.	[80]
China	662	n.r	n.r.	n.r	n.r.	115	18.5	n.r.	n.r.	60	9	20	3	28	4.2	6	0.9	[81]
China	1400	752	53.7	92	6.5	197	14	45	3.2	124	8.8	n.r.	n.r.	22	1.5	6	0.4	[82]
China	401	238	59.3	n.r.	n.r	n.r.	n.r.	n.r.	n.r.	n.r.	n.r.	n.r.	n.r.	n.r.	n.r.	n.r.	n.r.	[83]
East China	229	153	66.8	22	9.6	51	22.2	23	10	26	11.3	18	7.8	n.r.	n.r.	1	0.4	[84]
China	100	69	69	9	9	22	22	6	6	13	13	n.r.		2	2	1	1	[85]
Korea	218	147	67.4	31	14.2	40	18.3	n.r.	n.r.	18	8.2	9	4.1	11	5	1	0.4	[86]
Korea	507	319	65.3	65	13.3	104	21.3	n.r.	n.r.	60	12.3	15	3.1	22	4.5	6	1.2	[87]
Japan	442	270	61	38	9	n.r.	n.r.	34	7.6	69	15.6	n.r.	n.r.	25	5.4	n.r.	n.r.	[88]
Japan	1497	901	61	188	13	371	24.7	116	7.7	189	12.4	n.r.	n.r.	61	4	8	0.53	[89]
Iran	314	251	79.9	17	5.4	43	16.6	n.r.	n.r.	24	7.6	15	4.7	3	0.9	1	0.3	[90]
Singapore	145	95	65.5	14	9.6	31	21.3	n.r.	n.r.	18	12.4	8	5.5	4	2.7	1	0.6	[91]
Malaysia	35	27	77.1	2	5.7	n.r.	n.r.	2	5.7	3	8.5	n.r.	n.r.	n.r.	n.r.	2	5.7	[92]
India	88	65	73.8	n.r	n.r.	n.r.	n.r.	n.r.	n.r.	n.r.	n.r.	n.r.	n.r.	n.r.	n.r.	n.r.	n.r.	[93]
India	1660	1090	65.6	98	5.9	61	3.6	25	1.5	34	2	n.r.	n.r.	5	0.3	4	0.2	[94]
India	606	492	81.2	33	5.4	70	11.6	n.r.	n.r.	40	6.6	17	2.8	12	1.98	1	0.16	[95]
India	83	60	72.2	5	6.5	n.r.	n.r.	n.r.	n.r.	n.r.	n.r.	n.r.	n.r.	n.r.	n.r.	n.r.	n.r.	[96]
India	317	285	89.9	26	8.2	n.r.	n.r.	317	n.r.	n.r.	n.r.	n.r.	n.r.	n.r.	n.r.	n.r.	n.r.	[97]
India	961	642	66.8	55	5.7	101	10.5	n.r.	n.r.	n.r.	n.r.	n.r.	n.r.	n.r.	n.r.	n.r.	n.r.	[98]
Southern India	510	342	67.1	30	5.9	10	1.9	5	0.9	4	0.7	n.r.	n.r.	n.r.	n.r.	1	0.19	[99]
Sri Lanka	50	40	80	4	8	n.r.	n.r.	n.r.	n.r.	n.r.	n.r.	n.r.	n.r.	n.r.	n.r.	n.r.	n.r.	[100]
North Africa																		
Algeria	68	36	52.9	2	2.9	n.r.	n.r.	n.r.	n.r.	4	5.8	2	2.9	2	2.9	3	4.4	[71]
Egypt	152	78	51.3	n.r.	n.r.	n.r.	n.r.	n.r.	n.r.	n.r.	n.r.	n.r.	n.r.	n.r.	n.r.	n.r.	n.r.	[101]
Egypt	41	22	53.6	2	4.8	n.r.	n.r.	n.r.	n.r.	n.r.	n.r.	n.r.	n.r.	n.r.	n.r.	n.r.	n.r.	[102]
Egypt	100	55	55	n.r.	n.r.	n.r.	n.r.	n.r.	n.r.	n.r.	n.r.	n.r.	n.r.	n.r.	n.r.	n.r.	n.r.	[103]
Morocco	72	37	51.3	n.r.	n.r.	n.r.	n.r.	n.r.	n.r.	n.r.	n.r.	n.r.	n.r.	n.r.	n.r.	n.r.	n.r.	[104]
South Africa																		
South Africa	128	54	42.1	n.r.	n.r.	n.r.	n.r.	n.r.	n.r.	n.r.	n.r.	n.r.	n.r.	n.r.	n.r.	n.r.	n.r.	[105]
South Africa	261	90	34.4	34	13	3	1.1	n.r.	n.r.	n.r.	n.r.	n.r.	n.r.	n.r.	n.r.	n.r.	n.r.	[106]
North America																		
Canada	573	366	63.8	64	11	143	25	n.r.	n.r.	n.r.	n.r.	n.r.	n.r.	n.r.	n.r.	n.r.	n.r.	[107]
USA and Canada	436	256	79	23	7.1	45	13.9	n.r.	n.r.	n.r.	n.r.	n.r.	n.r.	n.r.	n.r.	n.r.	n.r.	[108]
USA	1014	460	45.3	112	11	442	43.5	113	10.1	256	25.2	n.r.		50	4.9	15	1.4	[65]
USA	68	45	66.1	4	5.8	12	17.6	n.r.	n.r.	9	13.2	2	2.9	n.r.	n.r.	3	2.9	[109]
Central-South America																		
Argentina	81 *	40	49.3	8	9.8	2	2.4	n.r.	n.r.	2		n.r.	n.r.	n.r.	n.r.	n.r.	n.r.	[110]
Colombia	69	40	58.8	10	4.5	11	15.9	n.r.	n.r.	8	11.6	n.r.	n.r.	3	4.3	n.r.	n.r.	[111]
Puerto Rico	141	56	66.7	2	2.4	n.r.	n.r.	n.r.	n.r.	n.r.	n.r.	n.r.	n.r.	n.r.	n.r.	n.r.	n.r.	[112]

* Females at risk were not included.

**Table 2 genes-14-00214-t002:** Summary of average frequencies of *DMD* gene variants identified in the seven considered macro-areas in the world.

World Macro-Areas	Suspected Dystrophinopathies	DMD/BMD Patients	Detection Rate	Large Deletions		Large Duplications		Point Mutations/Small Rearrangements	
	N	N		N	%	N	%	N	%
Western Europe	8922	7670	86.0	5011	65.3	991	12.9	1668	21.7
East Europe	826	620	75.1	367	59.2	75	12.1	178	28.7
Asia	12084	9913	82.0	7433	74.9	891	9.0	1589	16.0
North Africa	433	236	54.5	228	96.6	4	0.02	4	0.02
South Africa	389	178	45.8	144	80.9	34	19.1	n.a.	n.a.
North America	2091	1972	94.3	1127	57.2	203	10.3	642	32.5
Central-South America	210	169	80.4	136	80.5	20	11.8	13	7.7
Total or percentages	25.7	20.7	80.8	14.4	69.6	2218	10.7	4094	19.7
This report	750	719	95.8	534	71.2	73	10.1	112	15.6

## Data Availability

Data are available on request.

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
