# Peer review of "Spectrum of Genetic Variants in the Dystrophin Gene: A Single Centre Retrospective Analysis of 750 Duchenne and Becker Patients from Southern Italy"

_genes, 2023, doi:10.3390/genes14010214_

Round 1

Reviewer 1 Report

The conclusions are insufficient and are not derived from the work carried out. It is suggested to redraft conclusions referring to the work carried out.

Figures in black and white are difficult to see, please add patterns or change to colors.

Regarding Central and South America, there is data from Mexico and it is not included, please include or justify why some countries are included and others missing. 

The originality of the study is not evident in the conclusions.

Reviewer 2 Report

This is an interesting manuscript that discusses distribution of mutations in DMD gene occurring in different geographical areas. Although it is a well thought study, the manuscript suffers from major shortcomings that should be addressed before publishing. These are my comments on this manuscript- 

1.     The manuscript lacks a clear hypothesis and could benefit immensely by stating the objectives of the study clearly.  

2.     The language is difficult to understand with confusing sentences at many instances in the manuscript. For example, 

a)     The sentence “Deletions represent …North America” from line 28-30 in the abstract is not clear and is ambiguous. Should be replaced with easily understood sentence.

b)    Line 47-48, the sentence “Depending on whether these mutations allow the preservation of the DNA reading frame or not, a partial production or the total absence of the protein will occur” is not accurate and is ambiguous.  

c)     Line 100-101, the sentence “However, both boys with rapid and fatal evolution within 2 years of symptom onset, and older patients can be similarly affected” is unclear

3.     As an important objective of this study is to compare the gene variants from patient database across different geographical locations, a rigorous statistical analysis must be performed to compare the results. The methods do not show any statistical approach to compare the datasets. 

4.     Conventions must be followed in the manuscript. Please change 

a)     “KD” to “kDa” in line 44.

b)    “DYS gene” to “DMD gene” in line 276. 

c)     “CK” appears for the first time in the manuscript in line 74. Please use “creatine kinase (CK)” during the first use and then abbreviate for the remainder of the text.

d)    “NH2 domain” to “N-terminal actin binding domain (N-ABD)” in line 108.

e)    “Splicing alternative” to “alternative splicing” in line 111.

5.     Figures in the manuscript could be improved to make them easier to understand. For example- 

a)     In Figure 1, it is difficult to make out the difference between the colors for “duplications” and “no mutations”. A different color scheme or usage of shades like that used in figure 2 is advisable. 

b)    Figure 2 shows two panels, one behind the other. Please correct that.

c)     In figure 4, the Y-axis title could be kept similar to Figure 3 to maintain consistency. Figures 4-8 The X axis should also be displayed as displayed in Figure 3 (rotated font), to clearly see the exon numbers and for the sake of consistency. 

d)    Figure 9, the statistical significance of the differences between groups is missing. The comparisons that should be made for example are- Are large deletions in 7 areas any different from each other?

6.     There are some inconsistencies between text and Figures. For example-

a)     Line 169-170, “The most frequent hot spots of deletions were between exons E2-E11 and E45-E55”, But Figure 3A shows high frequency for deletions for exons-12-18 (at least higher than 45-55)

b)    Line 202-203, “Either single or multiple exon deletions prevalently occurred in one hot spot at the 3′ end (exons E44- E55) of the DMD gene (Figure 5).” But the Figure 5 legend says “Frequency of single exon deletions in the dystrophin gene in patients with DMD”

c)     It is unclear as to how the “frequency of occurrence” is calculated. Figure & shows the frequency of single exon deletion for exon number 45-48 to be higher than 120. But line 226-227 says “deletions of single exons were found in 25 patients (Figure 7)” It is unclear how the frequency of occurrence can be greater than the number of cases. Similarly for Figure 8, the total number of patients with single exon duplication is 4 while frequencies for occurrence in hotspot region is >5. 

7.     Dystrophin protein (muscle isoform) is composed of 3685 amino acids. Please correct it in line 44.

8.     Line 51-52, the sentence “Dystrophin acts as a bridge between the inner of the muscle fiber and the sarcolemma” is not correct. Dystrophin actually acts as a bridge between sarcolemma and actin cytoskeleton. Please correct the sentence. 

Round 2

Reviewer 1 Report

In the discussion, it stated that...

The reduced percentage of point mutations observed in the last decade in both DMD 384 and BMD patients should be related to several reasons: first, it is likely the result of the 385 genetic counseling and prenatal diagnosis, as previously mentioned; second, patients 386 without DMD gene deletions/duplications arrive late in tertiary centers and laboratories; and third, the diagnosis in BMD patients often occurs later in life.

How late diagnosis of BMD causes a reduction in point mutations? 

Delayed diagnosis is common in DMD, How many years of delayed diagnosis do you have? 

With regard to deletions/duplications, are they expected to decrease or increase over the years? why do you think you should put those in figures? Is it interesting in a way that is not yet clear to readers? please explain in detail 

What do you mean by the expression "however causative therapy is currently available only for patients with specific deletions or point mutations" What is causative therapy? please write differently or explain better.

Conclusions are still flawed,

You make several analyzes of mutation frequencies, between countries, in several years and only conclude that the diagnosis is better with NGS and MLPA (which we all already know).

Please write conclusions regarding the analyzes you did.

Reviewer 2 Report

The authors have provided a good account of the changes in the manuscript but unfortunately the overall impact of the study still remains low in spite of the changes. In the present form the manuscript does not provide any novel information. The language of the manuscript still suffers from major drawback, where the modified sentences are still ambiguous.
